# Calculation of a Climate Change Vulnerability Index for Nakdong Watersheds Considering Non-Point Pollution Sources

**Jungmin Kim * and Heongak Kwon**

Nakdong River Environment Research Center, National Institute of Environmental Research, 24-11, Gukgasan, dan-daero 52-gil, Guji-myeon, Dalseong-gun, Daegu 43008, Korea; hun7082@korea.kr
* Correspondence: jungminkim6214@gmail.com; Tel.: +82-56-602-2717

**Abstract:** As a response to climate change, South Korea has established its third National Climate Change Adaptation Plan (2021–2025) alongside the local governments' plans. In this study, proxy variables in 22 sub-watersheds of the Nakdong River, Korea were used to investigate climate exposure, sensitivity, adaptive capacity, and non-point pollution in sub-watersheds, a climate change vulnerability index (CCVI) was established, and the vulnerability of each sub-watershed in the Nakdong River was evaluated. Climate exposure was highest in the Nakdong Estuary sub-watershed (75.5–81.7) and lowest in the Geumhogang sub-watershed (21.1–28.1). Sensitivity was highest (55.7) in the Nakdong Miryang sub-watershed and lowest (19.6) in the Habcheon dam sub-watershed. Adaptive capacity and the resulting CCVI were highest in the Geumhogang sub-watershed (96.2 and 66.2–67.9, respectively) and lowest in the Wicheon sub-watershed (2.61 and 18.5–20.4, respectively), indicating low and high vulnerabilities to climate change, respectively. The study revealed that the high CCVI sensitivity was due to adaptive capacity. These findings can help establish rational climate change response plans for regional water resource management. To assess climate change vulnerability more accurately, regional bias can be prevented by considering various human factors, including resources, budget, and facilities.

**Keywords:** shared socioeconomic pathways; climate change vulnerability index; climate exposure; sensitivity; adaptive capacity

## 1. Introduction

Abnormal climate phenomena have been occurring with increased frequency worldwide, including on the Korean Peninsula, and a trend of increasing damage and influence associated with climate change has been observed in several fields of society with increasing climate uncertainty [1]. The Shared Socioeconomic Pathway (SSP) scenario from the Sixth Assessment Report of the Intergovernmental Panel on Climate Change (IPCC) forecasts that the average annual temperature and precipitation on the Korean Peninsula will increase by 7.0 °C and 14%, respectively, in the high-carbon scenario (SSP5-8.5) and by 2.6 °C and 3%, respectively, in the low carbon scenario (SSP1-2.6) by 2100. In particular, the maximum five-day precipitation and the number of top 5% extreme precipitation days are expected to increase; hence, responses to climate change are required [2] and the country must provide a quick, efficient, and effective response to climate change. South Korea has been establishing a National Climate Change Adaptation Plan based on an analysis of the impact and vulnerability of the country to climate change with the enactment of the Framework Act on Low Carbon Green Growth in April 2010. The establishment of the third National Climate Change Adaptation Plan (2021–2025) has been completed [3]. The term "climate change vulnerability" indicates how much systems such as the Earth, living organisms, and socioeconomic systems are susceptible to and unable to cope with the adverse effects of climate change [4,5]. The United Nations Development Program [4] considers climate

change vulnerability as a function of climate change sensitivity and adaptive capacity and expressed it as:

$$\text{Vulnerability} = f\,(\text{Sensitivity, Adaptive Capacity})$$

The climate change vulnerability index (CCVI), which is used to assess climate change vulnerability, is a useful tool for assessing the potential impact of climate change. Because climate exposure, sensitivity, and adaptive capacity cannot be measured directly, studies have used composite indicators to assess climate change vulnerability by selecting proxy variables to represent each parameter and then performing a standardization process for operation using the proxy variables [6–11]. The calculation of various indices using climate change scenarios and models has also been studied [12–15]. Several notable studies have investigated climate change vulnerability. Moss et al. [9] compared and analyzed the vulnerability to climate change for various countries by using 17 proxy variables to calculate vulnerability in terms of human/infrastructure, health/welfare, economy, and ecosystem. Al-Kalbani et al. [16] calculated the CCVI using the following four factors: water resources stress, water development pressure, ecological health, and management capacity, according to a manual presented by the United Nations Environment Program (UNEP). Edmonds et al. [11] calculated a new composite CCVI by estimating new weights in four vulnerable areas (ecosystem services, food, human habitat, health, infrastructure, and water) with data collected from over 100 countries by the University of Notre Dame. Leveque et al. [17] researched the hydrological and social impacts of climate change to examine the vulnerability of water sources in Quebec, Canada. Kim et al. [12] assessed the regional vulnerability of 14 major farmed species based on seawater temperature and salinity variation according to the AR5 representative concentration pathway (RCP) scenario. They investigated the farmed species and regional vulnerability by calculating a weighted vulnerability index after selecting two items to describe exposure and sensitivity and seven items to denote adaptive capacity. Noorissameleh et al. [15] used the standardized precipitation-evapotranspiration index (SPEI) and precipitation effectiveness variables (PEVs) extracted from the conjunctive precipitation effectiveness index (CPEI) to investigate the severity of droughts because of climate change. The authors also compared the magnitude of hydrological drought using the normalized differential water index (NDWI) and the streamflow draft index (SDI) according to the RCP climate change scenario. To examine cases of climate change vulnerability research for the establishment of climate change adaptation plans for local governments, Donggu District, Daejeon [18] assessed climate change vulnerability for health, disaster, agriculture and livestock industries, forest ecosystems, and water management and used the climate change scenario to establish detailed action plans. Gyeonggi-do [19] selected common evaluation indices for 31 municipalities and calculated the vulnerability index by applying weights derived through the analytical hierarchy process (AHP). Furthermore, Jeollabuk-do [20] assessed vulnerability by calculating vulnerability resilience indicators (VRIs) for water management, health, ecosystem, and industrial areas in 14 municipalities. There are many previous studies that evaluated the vulnerability of climate change in consideration of water quality factors, but few studies have used water quality factors for the purpose of non-point pollution. Therefore, a new water quality factor was calculated in consideration of basin characteristics and non-point pollution in this study.

In this study, a new water quality index was calculated using alternative evaluation indices that consider the ETCCDI, land use, load, and social infrastructures according to the AR6 SSP scenario presented by the IPCC, and the CCVI for each sub-watershed in the Nakdong River system was analyzed using this index.

## 2. Materials and Methods

### 2.1. Study Area

The Ministry of Environment establishes a Water Environment Management Plan that includes the objectives and directions of the national water environment management policy every decade, with Watershed Water Environment Management Master Plans for four major rivers (Han River, Nakdong River, Geum River, and Yeongsan River). The

Nakdong River system covers 32 sub-watersheds and 9 metropolitan areas and provinces, with 8 weirs constructed along the mainstream [21]. The targets of this study were the 22 sub-watersheds that lie along the mainstream of Nakdong River, as shown in Figure 1.

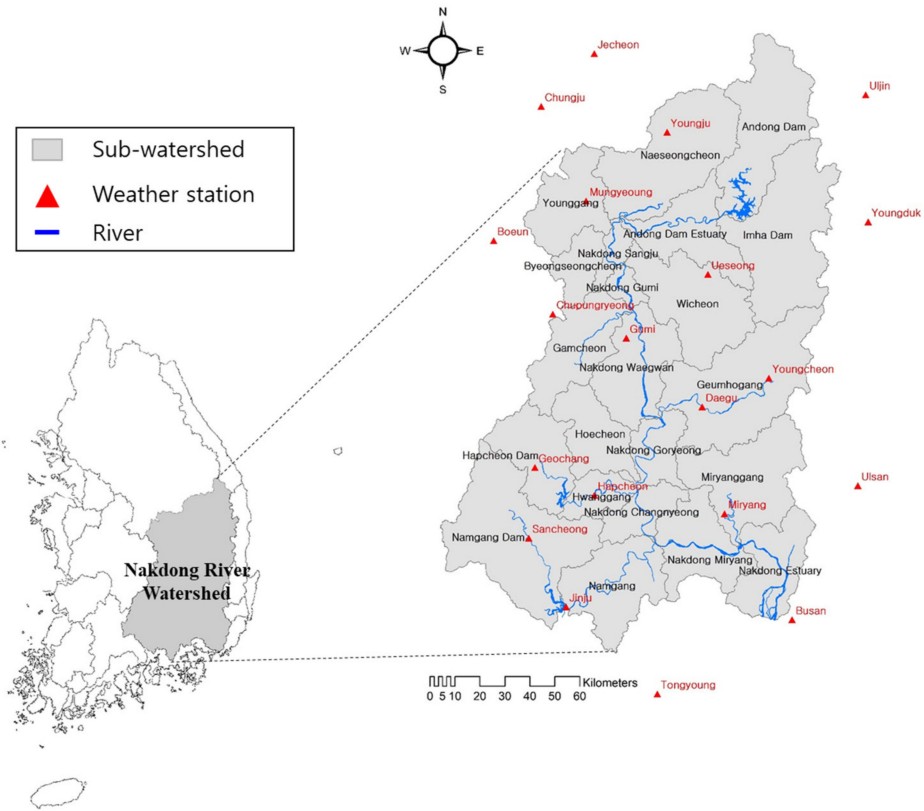

**Figure 1.** Study area (Nakdong River).

Studies related to the vulnerability of the Nakdong River have focused on floods and droughts [22–24]. An analytical study on the vulnerability of social infrastructures according to climate change [25], studies on the stability of water supply [26–28], and a study on the assessment of vulnerability related to weather events such as heatwaves [29] have also been conducted in the study area.

### 2.2. Climate Change Vulnerability Index

The CCVI can be calculated using three factors: how much a system is exposed to climate change (climate exposure), how much a system is sensitive to climate change (sensitivity), and the capacity of a system to adapt to climate change (adaptive capacity).

$$\text{Vulnerability} = (\alpha \times \text{climate exposure} + \beta \times \text{sensitivity}) - \gamma \times \text{adaptive capacity}$$

Climate exposure comprises elements such as temperature, precipitation, and relative humidity or other indices based on climate elements. Sensitivity, which indicates the sensitivity of a system to climate exposure, comprises the number and density of vulnerable objects. The impact of climate change can be expressed as a combination of climate exposure and sensitivity (Figure 2). Adaptive capacity indicates the associated policies and technical degree to which the impact of climate change can be reduced, and comprises the gross regional product, the number of fire department personnel, and the capacity of any sewage treatment facilities in an area. Lastly, $\alpha$, $\beta$, and $\gamma$ are the weights of each factor, and their sum must be equal to one. These weights can vary because of differences in the items used for vulnerability assessment [30].

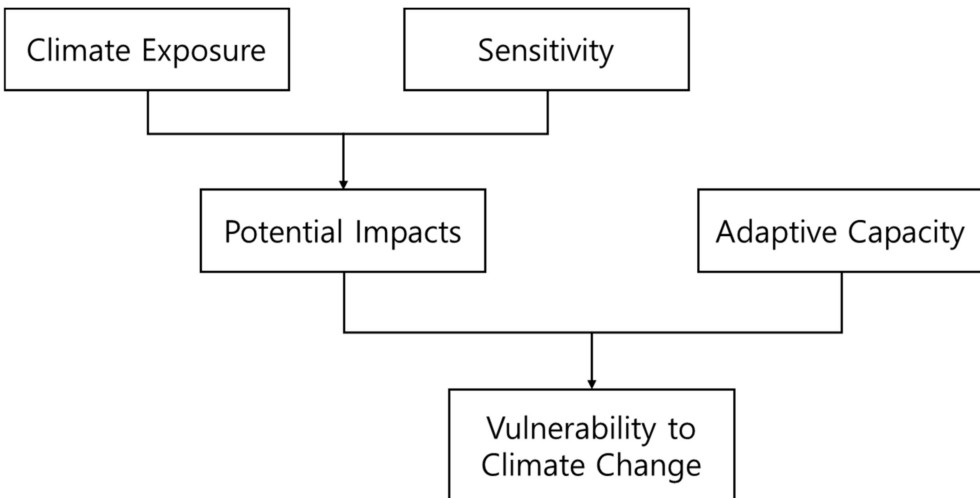

**Figure 2.** Schematic of climate change vulnerability [31].

The proxy variables describing climate exposure, sensitivity, and adaptive capacity were selected based on indices suggested by the Gyeonggi Research Institute [19], the Jeonbuk Development Institute [20], and the Donggu District, Daejeon [18], and concepts presented by the IPCC [5], UNDP [4], and Moss et al. [6]. The final proxy variables selected for the assessment of climate change vulnerability in this study are listed in Table 1. The CCVI was calculated by selecting seven proxy variables for climate exposure, four proxy variables for sensitivity, and two proxy variables for adaptive capacity.

**Table 1.** List of proxy variables that describe climate exposure, sensitivity, and adaptation capability.

| Category | Subtleties | ID | Proxy Variables |
|---|---|---|---|
| Exposure | Weather | PRCPTOT | Annual maximum one-day precipitation (PRCP) (mm) |
| | | R10mm | Annual count of days when PRCP $\geq$ 10 mm (days) |
| | | R80mm | Annual count of days when PRCP $\geq$ 80 mm (days) |
| | | CDD | Maximum number of consecutive days with daily PRCP < 1 mm (days) |
| | | CD5Day | Number of days without rain for over five consecutive days (day) |
| | | CD10Day50mm | Number of days of rainfall >50 mm after no rain for 10 days (event) |
| | | EDI | Effective drought index (EDI) |
| Sensitivity | Governance | | Pollution load discharged from non-point sources in watershed (kg/day) |
| | | | Number of pigs and cattle by area (ea/km$^2$) |
| | Environment | | Area ratio of paddies and upland field by sub-watershed (%) |
| | | | Curve number (CN) value |
| Adaptive capacity | Infrastructure | | Percentage of sewered population (%) |
| | | | Capacity of sewage treatment facilities (m$^3$) |

### 2.2.1. Standardization and Calculation of the Climate Change Vulnerability Index

To create one total vulnerability index from the selected individual vulnerability indices, multiple variables with different units and properties require aggregation and conversion. Methods such as normalization, weighting, and aggregation can convert data to a usable form [32]. Standardization methods include ranking, z-score, rescaling, distance to reference country, logarithmic transformation, categorical scale, indicators above or below the mean, cyclical indicators (OEDC), the balance of opinion (EC), and the percentage of annual differences over consecutive years [33]. In this study, the data were standardized

using rescaling, and the standard equations applied are shown in Table 2. The maximum and minimum values of the standardized index are 100 and 0, respectively.

**Table 2.** Standard equations used to develop the complex index describing vulnerability.

| Method | Equation |
|---|---|
| Raking | $I_{qc}^t = Rank\left(x_{qc}^t\right)$ |
| Z-score | $I_{qc}^t = \dfrac{x_{qc}^t - x_{qc=\bar{c}}^t}{\sigma_{qc=\bar{c}}^t}$ |
| Rescaling | $I_{qc}^t = \dfrac{x_{qc}^t - min_c\left(x_q^{t0}\right)}{max_c\left(x_q^{t0}\right) - min_c\left(x_q^{t0}\right)}$ |
| Distance to reference country | $I_{qc}^t = \dfrac{x_{qc}^t}{x_{qc=\bar{c}}^{t0}}$ or $I_{qc}^t = \dfrac{x_{qc}^t - x_{qc=\bar{c}}^{t0}}{x_{qc=\bar{c}}^{t0}}$ |
| Logarithmic transformation | $I_{qc}^t = ln\left(x_{qc}^t\right)$ |
| Categorical scale | If $x_{qc}^t$ is in the upper 5-percentile, then $y_{qc}^t = 100$<br>If $x_{qc}^t$ is in the upper 15-percentile, then $y_{qc}^t = 80$<br>If $x_{qc}^t$ is in the upper 35-percentile, then $y_{qc}^t = 60$ |
| Indicators above or below the mean | If $\dfrac{x_{qc}^t}{x_{qc=\bar{c}}^{t0}} > (1+p)$, then $I_{qc}^t = 1$<br>If $\dfrac{x_{qc}^t}{x_{qc=\bar{c}}^{t0}} > (1-p)$, then $I_{qc}^t = -1$<br>If $(1-p) < \dfrac{x_{qc}^t}{x_{qc=\bar{c}}^{t0}} < (1+p)$, then $I_{qc}^t = 0$ |
| Cyclical indicators (OECD) | $I_{qc}^t = \dfrac{x_{qc}^t - E_t\left(x_{qc}^t\right)}{E_t\left(\left|x_{qc}^t - E_t\left(x_{qc}^t\right)\right|\right)}$ |
| Balance of opinion (EC) | $I_{qc}^t = \dfrac{100}{N_e}\sum\limits_{e}^{N_e} sgn_e(x_{qc}^t - x_{qc}^{t-1})$ |
| Percentage of annual difference over consecutive years | $I_{qc}^t = \dfrac{x_{qc}^t - x_{qc}^{t-1}}{x_{qc}^t}$ |

$x_{qc}^t$ is the value of indicator $q$ for a region $c$ when the time is $t$. $\bar{c}$ is the reference country. The operator *sgn* indicates the sign of the argument (i.e., +1 if the argument is positive and −1 if the argument is negative). *Ne* is the total number of experts surveyed [33].

Yoo and Kim [34] stated that elective research for proxy variables is required to determine an appropriate vulnerability index for the 15 metropolitan areas and provinces in South Korea. Therefore, they presented a minimum CCVI dataset that is suitable for the situation in South Korea through the principal component analysis.

The CCVI was determined with the indices calculated for each indicator using Equation (1):

$$\text{CCVI} = \frac{Avg(Sensivity\ Index,\ Climate\ Exposure\ Index) + Adaptive\ Capacity\ Index}{2} \quad (1)$$

As illustrated in Figure 2, the potential impact is determined from a combination of climate exposure and sensitivity of a system using this equation, and the vulnerability index is determined by combining the potential impact with adaptive capacity. Moss et al. [6] and Yoo and Kim [34] defined the VRI, with a large VRI value implying that many positive numbers lead to elasticity and lower vulnerability. When a vulnerability is expressed as a positive number, a higher value implies that negative factors are more significant. This study also included the elasticity concept, implying that higher vulnerability values indicate lower CCVI values.

2.2.2. Data Used

KACE-1-0-G was used to describe the climate change scenario in terms of climate exposure. This general circulation model (GCM) was developed by the Korea Meteorological Administration as part of the AR6 SSP provided by the IPCC. Table 3 shows the explanation of AR6 climate change scenarios.

**Table 3.** Explain of AR6 climate change scenario from IPCC [35].

| AR6 SSP | Challenges | Illustrative Starting Points for Narratives |
|---|---|---|
| SSP1-2.6 | Low for mitigation and adaptation | Sustainable development proceeds at a reasonably high pace, inequalities are lessened, and technological change is rapid and directed toward environmentally friendly processes, including lower carbon energy sources and high productivity of land. |
| SSP2-4.5 | High for mitigation and adaptation | Unmitigated emissions are high due to moderate economic growth, a rapidly growing population, and slow technological change in the energy sector, making mitigation difficult. Investments in human capital are low, inequality is high, a regionalized world leads to reduced trade flows, and institutional development is unfavorable, leaving large numbers of people vulnerable to climate change and many parts of the world with low adaptive capacity. |
| SSP3-7.0 | High for adaptation, low for mitigation | A mixed world, with relatively rapid technological development in low carbon energy sources in key emitting regions, leading to relatively large mitigative capacity in places where it matters most to global emissions. However, in other regions development proceeds slowly, inequality remains high, and economies are relatively isolated, leaving these regions highly vulnerable to climate change with limited adaptive capacity. |
| SSP5-8.5 | High for mitigation, low for adaptation | In the absence of climate policies, energy demand is high and most of this demand is met with carbon-based fuels. Investments in alternative energy technologies are low, and there are few readily available options for mitigation. Nonetheless, economic development is relatively rapid and itself is driven by high investments in human capital. Improved human capital also produces a more equitable distribution of resources, stronger institutions, and slower population growth, leading to a less vulnerable world better able to adapt to climate impacts. |

Detailed data were produced for each meteorological station through statistical correction. We selected 16 meteorological stations with climate data covering 30 years (1981–2010) comprising the reproduction period in the Nakdong Geumho River watershed. Detailed data were produced for rainfall data for 2015–2100 using the empirical and simple quantile mapping techniques. The annual maximum one-day precipitation was defined as the amount of precipitation on the day with the highest daily precipitation in a particular year. The annual count of days when PRCP was ≥10 and 80 mm was defined as the number of days on which daily rainfall of ≥10 and 80 mm was observed.

The "days without rain" was defined as the number of days on which the observed rainfall was ≤1 mm. The maximum number of consecutive days with daily PRCP < 1 mm was determined by the maximum number of consecutive "days without rain." The number of days with rainfall >50 mm after no rain for 10 days was defined as the number of days on which rainfall of >50 mm was observed after 10 consecutive "days without rain." An effective drought index (EDI) was calculated to reflect the persistence and continuity of drought using climate change scenarios. The EDI calculates daily drought by comparing it with the average drought accumulated over a period of more than one year by considering the loss due to

runoff and evaporation from the available water resources that result from precipitation, which increases over time [36]. The EDI can be expressed using Equations (2)–(4):

$$EP_i = \sum_{n=1}^{i} \left[ \left( \sum_{m=1}^{n} P_m \right) / n \right] \tag{2}$$

$$DEP = EP - MEP \tag{3}$$

$$\text{EDI} = \frac{DEP}{ST(DEP)} \tag{4}$$

where effective precipitation (*EP*) indicates the cumulative effective precipitation for 365 days from a specific date; $P_m$ indicates the daily precipitation *m* days before a specific date; and *i* indicates the precipitation aggregation period, with a minimum value of 365 days. *n* indicates the duration of the summation of the precipitation. In addition, the mean effective precipitation (*MEP*) indicates the climatological mean *EP*, and the deviation effective precipitation (*DEP*) indicates the water surplus and scarcity in a specific period and space. The range of EDI is presented in Table 4. A negative value indicates a higher intensity of the drought.

**Table 4.** Categories associated with EDI values [37].

| EDI Value | Category |
|---|---|
| 2.00 or more | Extremely wet |
| 1.50 to 1.99 | Severely wet |
| 1.00 to 1.49 | Moderately wet |
| 0 to 0.99 | Mildly wet |
| −0.99 to 0 | Mild drought |
| −1.49 to −1.00 | Moderate drought |
| −1.99 to −1.50 | Severe drought |
| −2 or more | Extreme drought |

To derive climate change sensitivity, the value calculated from the total maximum daily load (TMDL) was used for the non-point source discharged pollution load (kg/day) by sub-watershed. The area ratio of paddy and upland field (%) was calculated using the land use map provided by the National Spatial Information Portal (www.nsdi.go.kr (accessed on 1 February 2020)). The curve number (CN) is an index indicating the ability of the basin to leak directly. CN was calculated using land use by the Environmental Geographic Information Service (EGIS) and soil maps provided by the Rural Development Administration, and data indicating the number of pigs and cattle by area were collected from the nationwide water pollution source survey. In terms of adaptive capacity, drainage system statistics were used for the capacity of sewage treatment facility (m$^3$) and the percentage of the sewered population (%). Data covering the 10 year period from 2010 to 2019 were used.

## 3. Results

### 3.1. Standardization of Climate Change Vulnerability Index

Standardization was performed after selecting 13 proxy variables from the three categories of climate exposure, sensitivity, and adaptive capacity. The data collected by each administrative district were reorganized by sub-watershed, and the proxy variables for each sub-watershed were determined using the rescaling method.

### 3.1.1. Exposure to Climate Change

Seven proxy variables were selected as indices to denote climate exposure. Six of the twenty-seven ETCCDI indices presented by the World Meteorological Organization (WMO) were selected by applying detailed data for KASE-1-0-G GCM provided by the Korea Meteorological Administration based on the SSP scenario of the Coupled Model

Intercomparison Project Phase 6 (CMIP6) by the IPCC. In addition, the drought condition was reflected by calculating the EDI. The standardized values of the proxy variables for the SSP-126, 245, 370, and 585 scenarios are shown in Tables 5–8, respectively.

**Table 5.** Standardized indices of climate change exposure in scenario SSP-126.

| Sub-Watershed | SSP-126 Scenario | | | | | | |
|---|---|---|---|---|---|---|---|
| | Rx1Day | CDD | CD5Day | R10mm | R50mm | CD10Day50mm | EDI |
| Andong Dam | 18.30 | 13.37 | 87.44 | 27.38 | 35.11 | 44.51 | 29.82 |
| Imha Dam | 9.41 | 36.12 | 100.00 | 5.91 | 17.38 | 42.87 | 85.04 |
| Andong Dam Estuary | 0.00 | 88.06 | 12.05 | 0.00 | 32.67 | 17.05 | 91.12 |
| Naeseongcheon | 31.59 | 65.60 | 0.00 | 38.98 | 69.41 | 46.38 | 32.39 |
| Younggang | 3.84 | 49.01 | 8.69 | 42.46 | 74.73 | 27.11 | 92.18 |
| Byeongseong-cheon | 35.59 | 0.00 | 49.41 | 46.34 | 42.25 | 23.58 | 52.13 |
| Wicheon | 2.26 | 91.11 | 21.33 | 1.38 | 17.84 | 8.63 | 82.96 |
| Nakdong Gumi | 20.45 | 71.42 | 45.62 | 22.84 | 4.38 | 2.84 | 100.00 |
| Gamcheon | 45.52 | 3.69 | 65.24 | 41.35 | 17.10 | 16.07 | 46.33 |
| Nakdong Waegwan | 21.16 | 73.17 | 41.42 | 23.44 | 4.94 | 3.23 | 87.86 |
| Nakdong Sangju | 8.08 | 55.09 | 18.91 | 39.92 | 60.17 | 23.57 | 94.40 |
| Geumhogang | 15.54 | 100.00 | 32.65 | 23.96 | 0.00 | 0.00 | 3.28 |
| Hoecheon | 55.17 | 70.50 | 33.52 | 60.30 | 61.95 | 42.68 | 26.72 |
| Nakdong Goryeong | 39.13 | 93.88 | 29.10 | 41.46 | 27.21 | 20.02 | 0.00 |
| Habcheon Dam | 49.46 | 42.03 | 45.02 | 72.31 | 75.63 | 48.96 | 16.47 |
| Hwanggang | 64.51 | 85.48 | 25.04 | 63.62 | 69.27 | 48.58 | 19.76 |
| Nakdong Changnyeong | 62.04 | 87.09 | 23.56 | 62.21 | 65.06 | 50.29 | 22.05 |
| Namgang Dam | 97.18 | 45.36 | 35.44 | 100.00 | 94.06 | 65.44 | 77.73 |
| Nakgang | 64.24 | 68.82 | 30.95 | 98.44 | 100.00 | 79.81 | 78.29 |
| Nakdong Miryang | 53.92 | 87.82 | 11.61 | 57.56 | 50.61 | 57.09 | 33.78 |
| Miryanggang | 45.05 | 92.25 | 24.36 | 49.42 | 36.50 | 41.70 | 27.35 |
| Nakdong Estuary | 100.00 | 55.01 | 52.29 | 92.74 | 94.59 | 100.00 | 33.90 |

**Table 6.** Standardized indices of climate change exposure in scenario SSP-245.

| Sub-Watershed | SSP-245 Scenario | | | | | | |
|---|---|---|---|---|---|---|---|
| | Rx1Day | CDD | CD5Day | R10mm | R50mm | CD10Day50mm | EDI |
| Andong Dam | 43.20 | 13.53 | 98.25 | 27.77 | 38.39 | 53.04 | 63.13 |
| Imha Dam | 22.73 | 35.30 | 100.00 | 6.24 | 18.45 | 41.37 | 58.93 |
| Andong Dam Estuary | 32.21 | 100.00 | 8.35 | 0.00 | 34.94 | 32.20 | 45.36 |
| Naeseongcheon | 67.27 | 69.35 | 2.59 | 37.75 | 77.50 | 81.62 | 100.00 |
| Younggang | 26.43 | 52.96 | 14.90 | 39.16 | 84.09 | 80.95 | 93.31 |
| Byeongseong-cheon | 32.76 | 0.00 | 55.22 | 44.61 | 44.53 | 50.25 | 47.38 |
| Wicheon | 20.25 | 97.21 | 20.61 | 2.31 | 18.41 | 18.83 | 37.37 |
| Nakdong Gumi | 4.91 | 68.75 | 54.58 | 25.52 | 3.63 | 24.38 | 100.00 |
| Gamcheon | 28.30 | 0.77 | 71.12 | 41.99 | 14.64 | 26.45 | 48.55 |
| Nakdong Waegwan | 5.90 | 68.85 | 49.57 | 26.07 | 4.18 | 23.00 | 89.12 |
| Nakdong Sangju | 19.02 | 57.02 | 26.11 | 37.63 | 67.17 | 70.79 | 93.84 |
| Geumhogang | 0.00 | 81.24 | 40.90 | 25.39 | 0.00 | 0.00 | 0.00 |
| Hoecheon | 46.85 | 59.13 | 28.65 | 60.69 | 57.69 | 46.88 | 33.77 |
| Nakdong Goryeong | 28.06 | 76.57 | 35.04 | 43.27 | 25.47 | 25.30 | 4.85 |
| Habcheon Dam | 40.59 | 34.99 | 42.04 | 71.18 | 70.75 | 51.72 | 93.90 |
| Hwanggang | 58.45 | 71.06 | 15.95 | 64.23 | 64.85 | 50.77 | 1.50 |
| Nakdong Changnyeong | 54.61 | 71.38 | 14.80 | 62.80 | 61.09 | 53.24 | 11.96 |
| Namgang Dam | 100.00 | 50.23 | 9.60 | 100.00 | 98.57 | 83.16 | 73.39 |
| Nakgang | 63.80 | 59.77 | 20.54 | 98.72 | 100.00 | 88.54 | 5.00 |
| Nakdong Miryang | 42.34 | 68.56 | 0.00 | 58.11 | 48.66 | 64.11 | 53.50 |
| Miryanggang | 32.13 | 73.26 | 20.04 | 49.96 | 34.90 | 45.70 | 39.43 |
| Nakdong Estuary | 96.68 | 57.37 | 27.23 | 93.37 | 93.28 | 100.00 | 86.50 |

**Table 7.** Standardized indices of climate change exposure in scenario SSP-370.

| Sub-Watershed | SSP-370 Scenario | | | | | | |
|---|---|---|---|---|---|---|---|
| | Rx1Day | CDD | CD5Day | R10mm | R50mm | CD10Day50mm | EDI |
| Andong Dam | 24.97 | 30.74 | 100.00 | 19.60 | 33.41 | 34.20 | 66.50 |
| Imha Dam | 9.96 | 59.31 | 97.27 | 0.76 | 16.53 | 52.74 | 32.84 |
| Andong Dam Estuary | 3.09 | 100.00 | 34.90 | 0.00 | 30.43 | 9.51 | 91.10 |
| Naeseongcheon | 51.35 | 70.54 | 0.00 | 34.82 | 70.28 | 49.60 | 32.19 |
| Younggang | 0.24 | 57.74 | 8.72 | 36.60 | 70.63 | 39.84 | 91.10 |
| Byeongseong-cheon | 41.45 | 3.68 | 47.71 | 44.73 | 40.64 | 16.46 | 47.74 |
| Wicheon | 0.00 | 96.30 | 44.51 | 4.75 | 15.77 | 2.33 | 93.80 |
| Nakdong Gumi | 13.97 | 62.15 | 72.49 | 32.28 | 2.17 | 8.19 | 100.00 |
| Gamcheon | 49.97 | 0.00 | 76.35 | 45.73 | 15.98 | 4.07 | 41.91 |
| Nakdong Waegwan | 14.08 | 63.34 | 63.37 | 32.67 | 2.97 | 8.56 | 98.46 |
| Nakdong Sangju | 4.23 | 59.83 | 22.23 | 37.21 | 56.52 | 34.38 | 93.89 |
| Geumhogang | 3.39 | 82.28 | 35.79 | 33.05 | 0.00 | 0.00 | 42.24 |
| Hoecheon | 45.02 | 60.41 | 58.48 | 65.23 | 57.41 | 50.14 | 15.99 |
| Nakdong Goryeong | 29.49 | 79.55 | 37.84 | 48.09 | 26.56 | 27.19 | 60.94 |
| Habcheon Dam | 38.09 | 35.61 | 75.71 | 74.60 | 69.25 | 42.10 | 14.07 |
| Hwanggang | 53.56 | 73.89 | 48.34 | 68.83 | 65.05 | 63.25 | 4.04 |
| Nakdong Changnyeong | 56.60 | 74.43 | 45.33 | 67.21 | 62.17 | 60.63 | 2.69 |
| Namgang Dam | 99.13 | 52.17 | 51.83 | 100.00 | 97.18 | 74.42 | 59.11 |
| Nakgang | 60.06 | 64.65 | 45.52 | 98.01 | 100.00 | 98.80 | 0.00 |
| Nakdong Miryang | 68.53 | 72.01 | 20.70 | 61.56 | 52.70 | 51.62 | 0.43 |
| Miryanggang | 51.81 | 76.78 | 35.97 | 54.82 | 38.13 | 37.37 | 9.07 |
| Nakdong Estuary | 100.00 | 55.59 | 54.99 | 97.66 | 97.45 | 100.00 | 63.68 |

**Table 8.** Standardized indices of climate change exposure in scenario SSP-585.

| Sub-Watershed | SSP-585 Scenario | | | | | | |
|---|---|---|---|---|---|---|---|
| | Rx1Day | CDD | CD5Day | R10mm | R50mm | CD10Day50mm | EDI |
| Andong Dam | 33.49 | 8.30 | 98.83 | 23.00 | 43.35 | 37.88 | 21.73 |
| Imha Dam | 11.86 | 34.41 | 100.00 | 2.18 | 23.38 | 31.86 | 58.10 |
| Andong Dam Estuary | 25.08 | 77.95 | 50.10 | 0.00 | 39.51 | 22.04 | 29.49 |
| Naeseongcheon | 67.94 | 57.17 | 41.90 | 37.35 | 78.10 | 56.46 | 25.49 |
| Younggang | 16.24 | 37.85 | 49.22 | 38.85 | 81.18 | 45.01 | 14.77 |
| Byeongseong-cheon | 40.43 | 0.00 | 72.41 | 43.13 | 46.35 | 28.45 | 13.76 |
| Wicheon | 14.75 | 83.92 | 46.33 | 2.38 | 23.04 | 12.61 | 21.27 |
| Nakdong Gumi | 6.62 | 71.37 | 45.17 | 23.49 | 8.42 | 12.12 | 1.31 |
| Gamcheon | 40.56 | 9.83 | 68.89 | 39.80 | 19.52 | 15.86 | 6.44 |
| Nakdong Waegwan | 7.42 | 72.98 | 40.38 | 24.24 | 8.56 | 11.88 | 2.32 |
| Nakdong Sangju | 12.88 | 44.90 | 50.56 | 37.11 | 66.13 | 39.14 | 15.14 |
| Geumhogang | 0.00 | 100.00 | 26.91 | 28.15 | 0.00 | 0.00 | 0.00 |
| Hoecheon | 47.54 | 69.08 | 22.08 | 59.09 | 58.62 | 40.03 | 65.87 |
| Nakdong Goryeong | 28.76 | 92.31 | 20.62 | 42.61 | 26.70 | 22.47 | 31.15 |
| Habcheon Dam | 39.05 | 45.31 | 23.48 | 70.87 | 72.79 | 43.81 | 9.68 |
| Hwanggang | 58.89 | 82.13 | 14.34 | 62.31 | 64.41 | 45.22 | 100.00 |
| Nakdong Changnyeong | 56.77 | 85.94 | 12.64 | 61.29 | 62.59 | 49.51 | 81.54 |
| Namgang Dam | 96.93 | 66.00 | 9.28 | 100.00 | 96.57 | 71.96 | 17.03 |
| Nakgang | 58.77 | 79.05 | 7.12 | 96.69 | 100.00 | 84.23 | 28.09 |
| Nakdong Miryang | 50.20 | 95.56 | 0.00 | 58.19 | 57.14 | 66.17 | 12.75 |
| Miryanggang | 38.24 | 98.08 | 13.49 | 51.12 | 40.95 | 47.28 | 11.60 |
| Nakdong Estuary | 100.00 | 64.41 | 37.90 | 95.85 | 98.77 | 100.00 | 75.18 |

The Nakdong Estuary sub-watershed showed the highest Rx1day values in scenarios SSP-125, 370, and 585. The Namgang Dam sub-watershed showed the highest value in the SSP-245 scenario. The sub-watershed that showed the lowest values were the same in scenarios SSP-245 and 585, with different values in other scenarios. The sub-watersheds associated with the Andong Dam Estuary, Geumhogang, and Wicheon showed the lowest

values in SSP-125, 245, 585, and 370. As for CDD, the Andong Dam Estuary showed the highest value in the scenarios SSP-245 and 370, and the Geumhogang sub-watershed showed the highest values in the scenarios SSP-126 and 585. For CD5day, the Imha Dam sub-watershed showed the highest value in the scenarios SSP-126, 245, and 585, and the Andong Dam sub-watershed showed the highest value in SSP-370. The sub-watersheds that showed the lowest value were Naeseongcheon in SSP-126 and 370, and Nakdong Miryang in the scenarios SSP-245 and 585. Regarding R10mm, the Namgang Dam sub-watershed showed the highest values and the Andong Dam Estuary sub-watershed showed the lowest values in all scenarios. For R80mm, the Namgang sub-watershed showed the highest values and the Geumhogang sub-watershed showed the lowest values in every scenario. For CD10Day50mm, the Nakdong Estuary sub-watershed showed the highest values and the Geumhogang sub-watershed showed the lowest values in every scenario. As for EDI, the Nakdong Gumi sub-watershed showed the highest values in scenarios SSP-126, 245, and 370, whereas the Hwanggang sub-watershed showed the highest value in the SSP-585 scenario. The Geumhogang sub-watershed showed the lower values in scenarios SSP-245 and 585, whereas the Nakdong Goryeong and Namgang sub-watersheds showed the lowest values in scenarios SSP-126 and 370, respectively. The higher the upstream of the Nakdong River, the higher the index when the rainfall was low, and the lower the downstream of the Nakdong River and Namgang watershed, the higher the index when the rainfall was high. The area midstream of Nakdong River and Geumhogang showed the lowest values in the index, including 50 mm or higher rainfall, indicating a low probability of occurrence of localized torrential rainfalls in this area.

### 3.1.2. Climate Change Sensitivity

In terms of the climate change sensitivity, the pollution load discharged from non-point sources in the watershed (kg/day), the area ratio of paddy and upland field by sub-watershed (%), number of pigs and cattle per area (ea/km$^2$), and the CN value were selected as proxy variables and standardized. The results of this standardization are outlined in Table 9.

**Table 9.** Standardized indices for climate change sensitivity in SSP scenarios.

| Sub-Watershed | Pollution Load Discharged from Non-Point Sources in Watershed | Number of Pigs and Cattle by Area | Area Ratio of Paddy and Upland Fields by Sub-Watershed | CN Value |
|---|---|---|---|---|
| Andong Dam | 37.34 | 0.00 | 42.62 | 19.47 |
| Imha Dam | 47.13 | 10.41 | 100.00 | 43.76 |
| Andong Dam Estuary | 30.51 | 52.03 | 1.40 | 49.36 |
| Naeseongcheon | 67.13 | 72.08 | 7.75 | 31.79 |
| Younggang | 26.39 | 26.41 | 12.23 | 49.89 |
| Byeongseong-cheon | 13.54 | 92.29 | 7.33 | 49.80 |
| Wicheon | 42.59 | 52.59 | 6.40 | 56.73 |
| Nakdong Gumi | 0.00 | 57.11 | 0.00 | 48.83 |
| Gamcheon | 34.22 | 58.29 | 3.65 | 40.85 |
| Nakdong Waegwan | 54.44 | 61.38 | 3.51 | 27.15 |
| Nakdong Sangju | 2.91 | 100.00 | 1.22 | 55.17 |
| Geumhogang | 100.00 | 35.23 | 8.69 | 61.19 |
| Hoecheon | 19.45 | 24.02 | 0.89 | 37.59 |
| Nakdong Goryeong | 29.69 | 51.22 | 4.64 | 58.59 |
| Habcheon Dam | 25.29 | 37.53 | 15.40 | 0.00 |
| Hwanggang | 8.42 | 38.55 | 4.85 | 54.98 |
| Nakdong Changnyeong | 13.24 | 59.44 | 2.50 | 100.00 |
| Namgang Dam | 72.15 | 23.02 | 11.71 | 34.88 |
| Nakgang | 51.85 | 62.40 | 4.94 | 89.73 |
| Nakdong Miryang | 47.30 | 95.90 | 0.87 | 78.67 |
| Miryanggang | 44.83 | 33.17 | 6.16 | 70.16 |
| Nakdong Estuary | 50.01 | 33.34 | 4.07 | 69.83 |

The pollution load discharged from non-point sources in the watershed was highest in the Geumhogang sub-watershed and lowest in the Nakdong Gumi sub-watershed. The area ratio of paddy and upland field by sub-watershed was highest in the Nakdong Sangju sub-watershed and lowest in the Andong Dam. The sub-watersheds that showed the highest and lowest values for the number of pigs and cattle by area were Imha Dam and Nakdong Gumi, respectively, and the sub-watersheds with the highest and lowest values of CN were Nakdong Changnyeong and Habcheon Dam, respectively.

### 3.1.3. Capacity for Climate Change Adaptivity

The capacity of sewage treatment facility ($m^3$/day) for each sub-watershed and the percentage of sewered population (%) were selected as proxy variables describing the capacity for climate change adaptivity. The data were collected and standardized, and the results are outlined in Table 10.

The capacity for sewage treatment was highest in the Geumhogang sub-watershed and lowest in the Nakdong Changnyeong sub-watershed. With the capacity for sewage treatment, the difference in the standardized index for each sub-watershed was large; this is because the facilities were often near industrial complexes. The percentage of the sewered population was highest in the Nakdong Estuary sub-watershed and lowest in the Hoecheon sub-watershed.

**Table 10.** Standardized indices of climate change sensitivity in SSP scenarios.

| Sub-Watershed | Capacity of Sewage Treatment Facilities | Percentage of Sewered Population |
|---|---|---|
| Andong Dam | 1.12 | 44.32 |
| Imha Dam | 0.23 | 29.04 |
| Andong Dam Estuary | 17.72 | 59.30 |
| Naeseongcheon | 2.14 | 42.47 |
| Younggang | 6.47 | 40.26 |
| Byeongseong-cheon | 41.00 | 80.26 |
| Wicheon | 1.13 | 4.10 |
| Nakdong Gumi | 0.10 | 38.24 |
| Gamcheon | 9.12 | 65.28 |
| Nakdong Waegwan | 71.57 | 76.63 |
| Nakdong Sangju | 0.98 | 28.17 |
| Geumhogang | 100.00 | 92.37 |
| Hoecheon | 19.52 | 0.00 |
| Nakdong Goryeong | 28.87 | 84.18 |
| Habcheon Dam | 4.65 | 33.33 |
| Hwanggang | 0.00 | 39.23 |
| Nakdong Changnyeong | 0.63 | 23.48 |
| Namgang Dam | 1.58 | 42.92 |
| Nakgang | 12.12 | 62.38 |
| Nakdong Miryang | 5.15 | 59.79 |
| Miryanggang | 2.98 | 53.96 |
| Nakdong Estuary | 17.43 | 100.00 |

### 3.2. Assessment of Climate Change Vulnerability

The CCVI for each scenario was derived using the climate change sensitivity and climate change adaptive capacity derived using the CCVI Equation (1).

### 3.2.1. SSP-126 Scenario

The CCVI was determined using the climate change exposure, sensitivity, and adaptive capacity from the sub-watershed calculations for the SSP-126 scenario. The results are presented in Table 11.

The highest climate change exposure was 75.50 in the Nakdong Estuary sub-watershed and the lowest was 25.06 in the Geumhogang sub-watershed. The highest climate change

sensitivity was 55.68 in the Nakdong Miryang sub-watershed, and the lowest was 19.56 in the Habcheon Dam sub-watershed. The highest climate change adaptive capacity was 96.18 in the Geumhogang sub-watershed and the lowest was 2.61 in the Wicheon sub-watershed. The highest CCVI was 68.18 in the Geumhogang sub-watershed, thus showing the highest climate change adaptive capacity, whereas the lowest was 19.25 in the Wicheon sub-watershed, thus showing the lowest climate change adaptive capacity. Therefore, this analysis suggested that the adaptive capacity significantly affects sensitivity because of its effect on the CCVI.

**Table 11.** Climate change vulnerability in the SSP-126 scenario.

| Sub-Watershed | Climate Change Exposure | Climate Change Sensitivity | Climate Change Adaptive Capacity | Climate Change Vulnerability Index |
|---|---|---|---|---|
| Andong Dam | 36.56 | 24.86 | 22.72 | 26.71 |
| Imha Dam | 42.39 | 50.32 | 14.63 | 30.50 |
| Andong Dam Estuary | 34.42 | 33.33 | 38.51 | 36.19 |
| Naeseongcheon | 40.62 | 44.69 | 22.31 | 32.48 |
| Younggang | 42.57 | 28.73 | 23.36 | 29.51 |
| Byeongseong-cheon | 35.62 | 40.74 | 60.63 | 49.40 |
| Wicheon | 32.22 | 39.58 | 2.61 | 19.25 |
| Nakdong Gumi | 38.22 | 26.48 | 19.17 | 25.76 |
| Gamcheon | 33.61 | 34.25 | 37.20 | 35.57 |
| Nakdong Waegwan | 36.46 | 36.62 | 74.10 | 55.32 |
| Nakdong Sangju | 42.88 | 39.83 | 14.57 | 27.96 |
| Geumhogang | 25.06 | 51.28 | 96.18 | 67.18 |
| Hoecheon | 50.12 | 20.49 | 9.76 | 22.53 |
| Nakdong Goryeong | 35.83 | 36.03 | 56.53 | 46.23 |
| Habcheon Dam | 49.98 | 19.56 | 18.99 | 26.88 |
| Hwanggang | 53.75 | 26.70 | 19.62 | 29.92 |
| Nakdong Changnyeong | 53.19 | 43.79 | 12.05 | 30.27 |
| Namgang Dam | 73.60 | 35.44 | 22.25 | 38.39 |
| Nakgang | 74.36 | 52.23 | 37.25 | 50.27 |
| Nakdong Miryang | 50.34 | 55.68 | 32.47 | 42.74 |
| Miryanggang | 45.23 | 38.58 | 28.47 | 35.19 |
| Nakdong Estuary | 75.50 | 39.31 | 58.72 | 58.06 |

3.2.2. SSP-245 Scenario

The CCVI was determined using the climate change exposure, sensitivity, and adaptive capacity by sub-watershed calculations for the SSP-245 scenario. The results are outlined in Table 12.

The highest climate change exposure in the SSP-245 scenario was 79.21 in the Nakdong Estuary sub-watershed, and the lowest was 21.08 in the Geumhogang sub-watershed. The highest climate change sensitivity was 55.68 in the Nakdong Miryang sub-watershed and the lowest was 19.56 in the Habcheon Dam sub-watershed. As in the SSP-126 scenario, the highest climate change adaptive capacity was 96.18 in the Geumhogang sub-watershed and the lowest was 2.61 in the Wicheon sub-watershed. Moreover, the highest CCVI was 66.18 in the Geumhogang sub-watershed and the lowest was 18.88 in the Wicheon sub-watershed.

**Table 12.** Climate change vulnerability in the SSP-245 scenario.

| Sub-Watershed | Climate Change Exposure | Climate Change Sensitivity | Climate Change Adaptive Capacity | Climate Change Vulnerability Index |
|---|---|---|---|---|
| Andong Dam | 48.19 | 24.86 | 22.72 | 29.62 |
| Imha Dam | 40.43 | 50.32 | 14.63 | 30.01 |
| Andong Dam Estuary | 36.15 | 33.33 | 38.51 | 36.63 |
| Naeseongcheon | 62.30 | 44.69 | 22.31 | 37.90 |
| Younggang | 55.97 | 28.73 | 23.36 | 32.86 |
| Byeongseong-cheon | 39.25 | 40.74 | 60.63 | 50.31 |
| Wicheon | 30.71 | 39.58 | 2.61 | 18.88 |
| Nakdong Gumi | 40.25 | 26.48 | 19.17 | 26.27 |
| Gamcheon | 33.12 | 34.25 | 37.20 | 35.44 |
| Nakdong Waegwan | 38.10 | 36.62 | 74.10 | 55.73 |
| Nakdong Sangju | 53.08 | 39.83 | 14.57 | 30.51 |
| Geumhogang | 21.08 | 51.28 | 96.18 | 66.18 |
| Hoecheon | 47.66 | 20.49 | 9.76 | 21.92 |
| Nakdong Goryeong | 34.08 | 36.03 | 56.53 | 45.79 |
| Habcheon Dam | 57.88 | 19.56 | 18.99 | 28.85 |
| Hwanggang | 46.69 | 26.70 | 19.62 | 28.15 |
| Nakdong Changnyeong | 47.12 | 43.79 | 12.05 | 28.75 |
| Namgang Dam | 73.56 | 35.44 | 22.25 | 38.38 |
| Nakgang | 62.34 | 52.23 | 37.25 | 47.27 |
| Nakdong Miryang | 47.90 | 55.68 | 32.47 | 42.13 |
| Miryanggang | 42.20 | 38.58 | 28.47 | 34.43 |
| Nakdong Estuary | 79.21 | 39.31 | 58.72 | 58.99 |

### 3.2.3. SSP-370 Scenario

The CCVI was determined using the climate change exposure, sensitivity, and adaptive capacity by sub-watershed calculated for the SSP-370 scenario. The results are shown in Table 13.

**Table 13.** Climate change vulnerability in the SSP-370 scenario.

| Sub-Watershed | Climate Change Exposure | Climate Change Sensitivity | Climate Change Adaptive Capacity | Climate Change Vulnerability Index |
|---|---|---|---|---|
| Andong Dam | 44.20 | 24.86 | 22.72 | 28.62 |
| Imha Dam | 38.49 | 50.32 | 14.63 | 29.52 |
| Andong Dam Estuary | 38.43 | 33.33 | 38.51 | 37.20 |
| Naeseongcheon | 44.11 | 44.69 | 22.31 | 33.35 |
| Younggang | 43.55 | 28.73 | 23.36 | 29.75 |
| Byeongseong-cheon | 34.63 | 40.74 | 60.63 | 49.16 |
| Wicheon | 36.78 | 39.58 | 2.61 | 20.40 |
| Nakdong Gumi | 41.61 | 26.48 | 19.17 | 26.61 |
| Gamcheon | 33.43 | 34.25 | 37.20 | 35.52 |
| Nakdong Waegwan | 40.49 | 36.62 | 74.10 | 56.33 |
| Nakdong Sangju | 44.04 | 39.83 | 14.57 | 28.25 |
| Geumhogang | 28.11 | 51.28 | 96.18 | 67.94 |
| Hoecheon | 50.38 | 20.49 | 9.76 | 22.60 |
| Nakdong Goryeong | 44.24 | 36.03 | 56.53 | 48.33 |
| Habcheon Dam | 49.92 | 19.56 | 18.99 | 26.86 |
| Hwanggang | 53.85 | 26.70 | 19.62 | 29.95 |
| Nakdong Changnyeong | 52.72 | 43.79 | 12.05 | 30.15 |
| Namgang Dam | 76.26 | 35.44 | 22.25 | 39.05 |
| Nakgang | 66.72 | 52.23 | 37.25 | 48.36 |
| Nakdong Miryang | 46.79 | 55.68 | 32.47 | 41.85 |
| Miryanggang | 43.42 | 38.58 | 28.47 | 34.73 |
| Nakdong Estuary | 81.34 | 39.31 | 58.72 | 59.52 |

The highest climate change exposure in the SSP-370 scenario was 81.34 in the Nakdong Estuary sub-watershed and the lowest was 28.11 in the Geumhogang sub-watershed. Similarly, the highest CCVI was 66.18 in the Geumhogang sub-watershed and the lowest was 18.88 in the Wicheon sub-watershed.

### 3.2.4. SSP-585 Scenario

The CCVI was determined using the climate change exposure and the sensitivity and adaptive capacity from the sub-watershed calculations for the SSP-585 scenario. The results are presented in Table 14.

The highest climate change exposure for the SSP-585 scenario was 81.73 in the Nakdong Estuary sub-watershed and the lowest was 22.15 in the Geumhogang sub-watershed. Similarly, the highest CCVI was 66.45 in the Geumhogang sub-watershed and the lowest was 18.50 in the Wicheon sub-watershed.

**Table 14.** Climate change vulnerability in the SSP-585 scenario.

| Sub-Watershed | Climate Change Exposure | Climate Change Sensitivity | Climate Change Adaptive Capacity | Climate Change Vulnerability Index |
|---|---|---|---|---|
| Andong Dam | 38.08 | 24.86 | 22.72 | 27.09 |
| Imha Dam | 37.40 | 50.32 | 14.63 | 29.25 |
| Andong Dam Estuary | 34.88 | 33.33 | 38.51 | 36.31 |
| Naeseongcheon | 52.06 | 44.69 | 22.31 | 35.34 |
| Younggang | 40.45 | 28.73 | 23.36 | 28.98 |
| Byeongseong-cheon | 34.93 | 40.74 | 60.63 | 49.23 |
| Wicheon | 29.18 | 39.58 | 2.61 | 18.50 |
| Nakdong Gumi | 24.07 | 26.48 | 19.17 | 22.22 |
| Gamcheon | 28.70 | 34.25 | 37.20 | 34.34 |
| Nakdong Waegwan | 23.97 | 36.62 | 74.10 | 52.20 |
| Nakdong Sangju | 37.98 | 39.83 | 14.57 | 26.74 |
| Geumhogang | 22.15 | 51.28 | 96.18 | 66.45 |
| Hoecheon | 51.76 | 20.49 | 9.76 | 22.94 |
| Nakdong Goryeong | 37.81 | 36.03 | 56.53 | 46.72 |
| Habcheon Dam | 43.57 | 19.56 | 18.99 | 25.28 |
| Hwanggang | 61.04 | 26.70 | 19.62 | 31.74 |
| Nakdong Changnyeong | 58.61 | 43.79 | 12.05 | 31.63 |
| Namgang Dam | 65.40 | 35.44 | 22.25 | 36.33 |
| Nakgang | 64.85 | 52.23 | 37.25 | 47.89 |
| Nakdong Miryang | 48.57 | 55.68 | 32.47 | 42.30 |
| Miryanggang | 42.97 | 38.58 | 28.47 | 34.62 |
| Nakdong Estuary | 81.73 | 39.31 | 58.72 | 59.62 |

The CCVI results by sub-watershed according to each scenario are shown in Figure 3. The spatial distributions of the CCVI by scenario were similar because the sensitivity and adaptive capacity did not only change under changes in the climate exposure according to the climate change scenario. The CCVI was high in the Naeseongcheon sub-watershed upstream of the Nakdong River, the Geumhogang sub-watershed in the midstream of the Nakdong River, and the Namgang and estuary sub-watersheds in the downstream of the Nakdong River. The CCVI was low in the Andong Dam, Imha Dam, Habcheon Dam, and the Wicheon sub-watersheds in upstream and midstream Nakdong River.

The high CCVI observed in the Geumhogang sub-watershed might be because of its low exposure to climate change and its high adaptive capacity that results from the high percentage of sewered population and the capacity for sewage treatment owing to urbanization. Although the highest climate exposure was observed in the estuary, its CCVI was high due to the low sensitivity and high adaptive capacity that result from urbanization, as with the Geumhogang sub-watershed. Low CCVIs were derived for the Wicheon, Hoecheon, Nakdong Gumi, and Hwanggang sub-watersheds because of the low percentage of sewered population and low capacity for sewage treatment in these areas.

The sensitivity for each SSP scenario associated with each proxy variable for the Geumhogang sub-watershed, and the Wicheon sub-watershed, with the highest and lowest CCVI, respectively, in every scenario, is represented graphically in Figure 4.

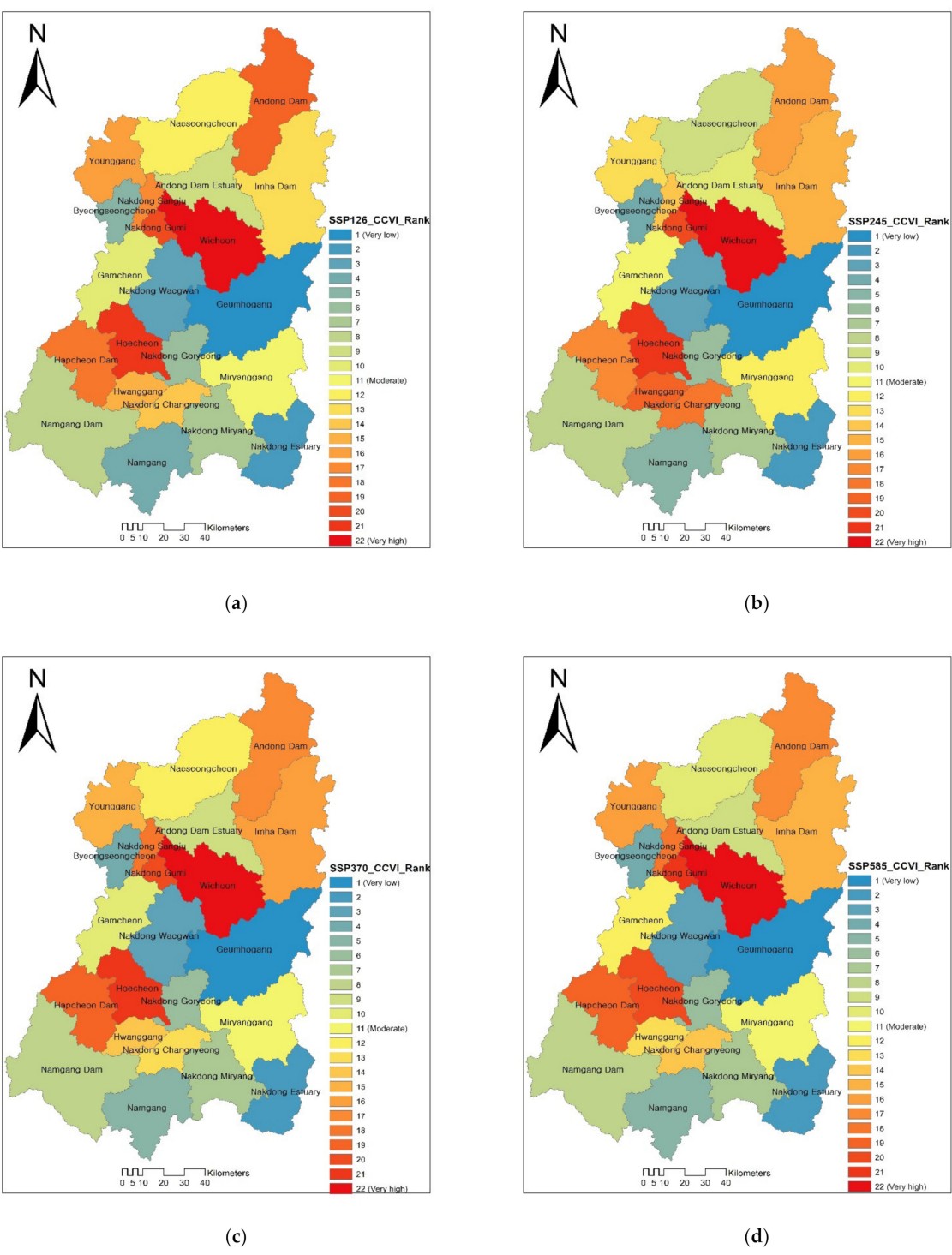

**Figure 3.** Spatial distribution of CCVI by Nakdong River sub-watershed (the closer to blue, the higher the CCVI, the lower the vulnerability to climate change; and the closer to red, the lower the CCVI and the higher the vulnerability to climate change). (**a**) ssp-126, (**b**) ssp245, (**c**) ssp-370, and (**d**) ssp-585.

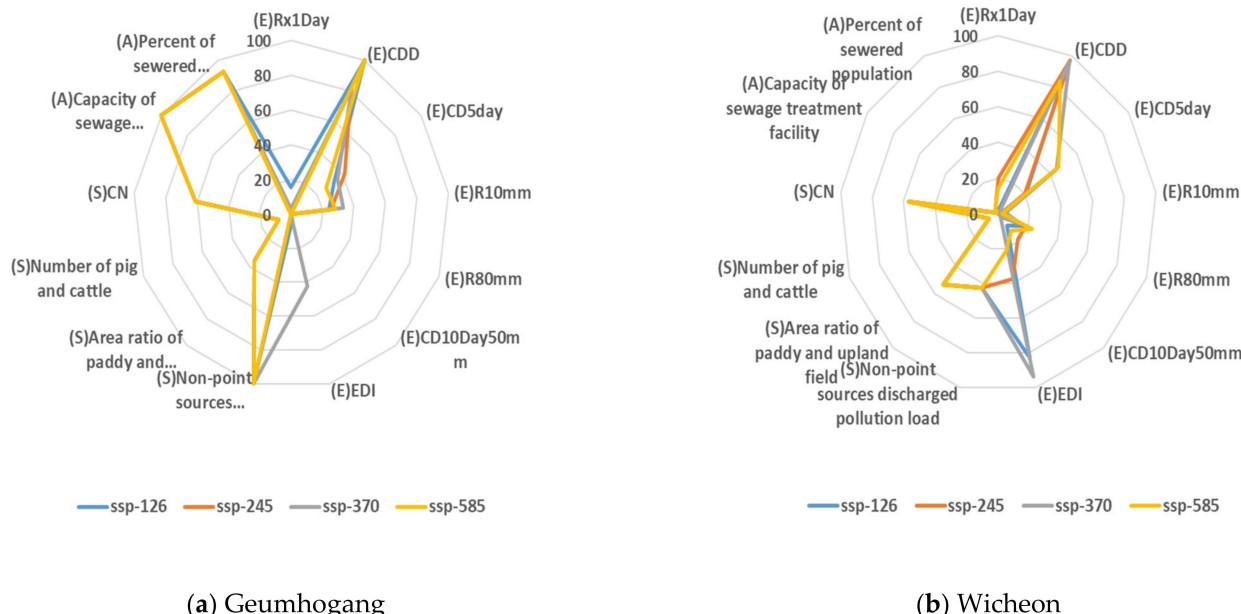

(**a**) Geumhogang                                     (**b**) Wicheon

**Figure 4.** Analysis of sensitivity by proxy variable for each scenario ((**a**) is the Geumhogang sub-watershed with the highest CCVI and (**b**) is the Wicheon sub-watershed with the lowest CCVI).

As shown in Figure 4, the CCVI became lower (vulnerability increases) as (E) and (S) extended closer to 100, and the CCVI became lower as (A) became closer to 0. Because of analyzing the proxy variable sensitivity for each scenario in the Geumhogang sub-watershed, with the highest CCVI in every scenario, the proxy variables CDD in the climate change exposure and non-point source discharged pollution load in the climate change sensitivity were high. However, the capacity for sewage treatment and the percentage of sewered population in the climate change adaptive capacity were high, resulting in a high CCVI and indicating low climate change vulnerability. The proxy variables CDD and EDI in the climate change exposure were high in the Wicheon sub-watershed, whereas the standard proxy values describing the adaptive capacity for climate change were lowest, resulting in a low CCVI and indicating a high climate change vulnerability. When the results were analyzed in association with the climate change scenario, the scenarios SSP-126 and SSP-370 showed similar results, whereas scenarios SSP-245 and SSP-585 showed similar results. Particularly for the Wicheon sub-watershed, the EDI was high in the SSP-125 and SSP-370 scenarios, and the analysis suggested that the increased number of days without rainfall resulted in a drought.

## 4. Conclusions

This study calculated a CCVI using alternative evaluation indices that considered the ETCCDI, land, and social infrastructures using the AR6 SSP scenarios provided by the IPCC to assess the non-point pollution vulnerability of the Nakdong River sub-watersheds while responding to climate change scenarios that are expected to occur. Detailed data were produced in line with the meteorological stations selected for use with KACE-1-0-G GCM, which was provided by the Korea Meteorological Administration. Seven ETCCDIs related to rainfall were calculated and selected as proxy variables describing climate exposure. In addition, four items considering soil, land use, and load were selected and set as proxy variables for sensitivity. Furthermore, two other parameters—the percentage of sewered population and the capacity for sewage treatment—were selected as proxy variables for adaptive capacity. Thus, the CCVI was calculated using 13 proxy variables. Variables with different units were standardized using the rescaling method. Table 15 shows the differentiation from previous studies.

**Table 15.** Comparing the precision of this study and previous studies.

| | Previous Study | This Study |
|---|---|---|
| 1 | No assessment of vulnerability to climate change targeting non-point pollution (health, disaster, forest, water management, water quality, ecosystem, industrial, food, etc.) | Assessment of vulnerability to climate change targeting non-point pollution (non-point pollution) |
| 2 | Selecting items for evaluating climate change vulnerability using the most data that can be collected (44 items; Hoque et al., 2022 33 items; Yoo and Kim, 2008 35 items; Ko and Kim, 2009) | Selection the minimum item required for amateur use (13 items) |
| 3 | When calculating climate change vulnerability indices, drought index was not included or used SPI and PDSI that used monthly data. (unit: 3, 6, 9, 12 monthly, etc.) | Use of EDI as drought index to reflect continuity and persistence of drought (unit: daily) |
| 4 | Selection of administrative districts as analysis areas (administrative districts; nation or metropolitan and state) | Selection of sub-watershed as analysis areas (22 sub-watersheds in Nakdong River; watershed or basin) |
| 5 | Use of AR4 or AR5 scenario from IPCC (AR4; SRES A2/B2/A1B AR5; RCP2.6/4.5/6.0/8.5) | Use of AR6 scenario from IPCC (SSP 1-2.6/2-4.5/3-7.0/5-8.5) |

When the standardized climate exposure was examined under every climate change scenario, even within the Nakdong River, Rx1day, R10mm, and R50mm were high in the Namgang Dam and Nakdong Estuary sub-watersheds, which are located in a rainy region due to the influence of Mount Jiri and the ocean. In contrast, the Wicheon, Gamcheon, and Geumhogang sub-watersheds, where the above-mentioned variables were low, showed high CDD and CD5day. These sub-watersheds are affected by their location in the upstream and midstream Nakdong River and the topography of mountains on the right side of the Korean Peninsula. Moreover, the Naeseongcheon and Younggang sub-watersheds showed high CDD and CD5day, but their R50mm was also high, indicating a high possibility of localized torrential rainfall during a rain event. The CD10Day50mm was high among the sub-watersheds located downstream of the Nakdong River, but the lack of rainfall is not expected to lead to drought if this were to occur in the midstream of the Nakdong River. The same pattern was also observed for the EDI. Similar to CDD, the EDI was high in the sub-watersheds located in the upstream and midstream areas of Nakdong River. In terms of climate change sensitivity, the non-point source discharged pollution load in each watershed, the number of pigs and cattle by area, and the area ratio of paddies and upland fields per sub-watershed were high in the tributary watersheds upstream of the Nakdong River with many livestock farms, farmlands, and sub-watersheds located in the middle and lower regions of Nakdong River. Regarding the adaptive capacity, the capacity for sewage treatment and the percentage of sewered population were high in the specific sub-watersheds with large cities such as Daegu, Busan, and Jinju, or industrial complexes. The calculation result of CCVI for each climate change scenario showed that in every scenario, the Geumhogang sub-watershed with the highest adaptive capacity showed the highest CCVI, and the Wicheon sub-watershed with the lowest adaptive capacity showed the lowest CCVI. The CCVIs of Imha Dam, Naeseongcheon, and adjacent sub-watersheds located upstream of the Nakdong River, Habcheon Dam, and Hwanggang sub-watersheds changed in accordance with the climate change scenario. However, the CCVIs were high for Geumhogang and adjacent sub-watersheds in the middle of the Nakdong River and sub-watersheds located in the lower reaches of the Nakdong River, including the Namgang sub-watershed. In contrast, the CCVIs were low in the Wicheon, Hoecheon, and Andong Dam sub-watersheds, which showed low adaptive capacity. Therefore, this study found that high CCVI sensitivity was due to adaptive capacity.

The findings of this study are expected to contribute to the establishment of rational climate change response plans for water resource management by quantitatively presenting the effects of climate change on the water resources and water quality management in the study area. Kim et al. [38] suggested that socioeconomic scenarios should be estimated and applied for a more accurate diagnosis of climate change responses. Therefore, additional research is necessary for consideration of human factors such as resources and budget as well as facilities to consider the equity of adaptive capacity by selecting factors (such as the percentage of sewered population and the capacity of sewage treatment facilities), which are inevitably unequally distributed in specific areas, as proxy variables.

**Author Contributions:** Conceptualization, J.K.; methodology, J.K. and H.K.; formal analysis, J.K. and H.K.; investigation, J.K. and H.K.; resources, J.K. and H.K.; data curation, J.K. and H.K.; writing—original draft preparation, J.K.; writing—review and editing, J.K.; visualization, J.K.; supervision, J.K.; project administration, J.K.; funding acquisition, H.K. All authors have read and agreed to the published version of the manuscript.

**Funding:** This work was supported by the National Institute of Environmental Research (NIER), funded by the Ministry of Environment (ME) of the Republic of Korea (grant number NIER-2021-01-01-042).

**Institutional Review Board Statement:** Not applicable.

**Informed Consent Statement:** Not applicable.

**Data Availability Statement:** Data are contained within the article.

**Conflicts of Interest:** The authors declare no conflict of interest.

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
