# Peer review of "Calculation of a Climate Change Vulnerability Index for Nakdong Watersheds Considering Non-Point Pollution Sources"

_applsci, doi:10.3390/app12094775_

Round 1
Reviewer 1 Report
Line85, the problem of present CCVI was suggested to be added to this paragraph, which is the reason that you propose the new water quality index.
Line 214, what are SSP-126, 245, 214 370, and 585 scenarios and their differences? Please explain them clearly.
Why not select the best scheme for evaluation?
Line 250, How does the CN value calculated? What does it mean physically?
Figure 3, please add names of different colored areas.
Line 347, how does the Figure 4 make and what's the meaning of the figure? Please explain it in details.
Section conclusions, what's the new contribution of your indices? Please add a table comparing the precision of yours and the traditional ones.
Reviewer 2 Report
I think that the research needs a statistical analysis of the data of the areas under study References need to be recent 2021 and 2022 for previous or similar studies The conclusion section needs to be shortened and a dissection section should be added to analyze and compare the results The research needs more visualization in displaying and analyzing the resultsAuthor Response
Please see the attachment.

Reviewer 3 Report
It is a very important issue on climate change field. The findings implicated to the local policy concern and plan for action to develop adaptation strategy.
Congratulations and good luck.
Reviewer 4 Report
The aim, method, and result have been presented well. But I have two major points about the selection Proxy variables and EDI.
- Why did you select proxy (Table 1)? You should present a reasonable reason for them.
- Why did you use EDI? What about other drought indexes?
The result without addressing the reason for variables selection (1,2) are under question.
Round 2
Reviewer 1 Report
Please see the blue words in the attachment.

Reviewer 2 Report
Accept in present form
Author Response
Thank you for your opinion.
Reviewer 4 Report
The authors's response is Ok.
Author Response
Thank you for your opinion.
Round 3
Reviewer 1 Report
Please see the blue words in the attachment.

Author Response
Thank you for your opinion. Please refer to the attached file.
